# Unconventional supercurrent phase in Ising superconductor Josephson junction with atomically thin magnetic insulator

H. Idzuchi[1,7], F. Pientka [1,2], K.-F. Huang[1], K. Harada[3], Ö. Gül [1], Y. J. Shin [1,8], L. T. Nguyen[4], N. H. Jo[5,6], D. Shindo[3], R. J. Cava[4], P. C. Canfield[5,6] & P. Kim [1✉]

In two-dimensional (2D) NbSe$_2$ crystal, which lacks inversion symmetry, strong spin-orbit coupling aligns the spins of Cooper pairs to the orbital valleys, forming Ising Cooper pairs (ICPs). The unusual spin texture of ICPs can be further modulated by introducing magnetic exchange. Here, we report unconventional supercurrent phase in van der Waals heterostructure Josephson junctions (JJs) that couples NbSe$_2$ ICPs across an atomically thin magnetic insulator (MI) Cr$_2$Ge$_2$Te$_6$. By constructing a superconducting quantum interference device (SQUID), we measure the phase of the transferred Cooper pairs in the MI JJ. We demonstrate a doubly degenerate nontrivial JJ phase ($\phi$), formed by momentum-conserving tunneling of ICPs across magnetic domains in the barrier. The doubly degenerate ground states in MI JJs provide a two-level quantum system that can be utilized as a new dissipationless component for superconducting quantum devices. Our work boosts the study of various superconducting states with spin-orbit coupling, opening up an avenue to designing new superconducting phase-controlled quantum electronic devices.

[1] Department of Physics, Harvard University, Cambridge, MA, USA. [2] Institut für Theoretische Physik, Goethe-Universität, Frankfurt am Main, Germany. [3] Center for Emergent Matter Science (CEMS), RIKEN, Hatoyama, Saitama, Japan. [4] Department of Chemistry, Princeton University, Princeton, NJ, USA. [5] Department of Physics and Astronomy, Iowa State University, Ames, IA, USA. [6] Ames Laboratory, Iowa State University, Ames, IA, USA. [7] Present address: WPI Advanced Institute for Materials Research and Center for Science and Innovation in Spintronics, Tohoku University, Sendai, Japan. [8] Present address: Center for Functional Nanomaterials, Brookhaven National Laboratory, Upton, NY, USA. ✉email: pkim@physics.harvard.edu

In a crystal which lacks inversion symmetry, the relativistic coupling between spins and electron orbits creates a momentum-dependent spin splitting, leading to spin-polarization without magnetism[1–4]. Two-dimensional transition metal dichalcogenides (TMDs), such as 2H phase of $NbSe_2$, $MoS_2$, and $TaS_2$, exhibit unusual superconducting properties stemming from strong spin–orbit coupling (SOC) in combination with their 2D structure with broken inversion symmetry. Several notable properties include in-plane upper critical fields that far exceed the paramagnetic spin limit of Bardeen–Cooper–Schrieffer (BCS) theory[5–7], co-existance of charge-density-waves with superconductivity down to monolayer limit[8], and higher order paramagnetic-limited superconductor–normal metal transitions[9]. These unusual properties result from an out-of-plane alignment of electron spins forming ICPs[5–9]. Early theoretical studies predicted anomalous Josephson coupling between two non-centrosymmeteric superconductors, which can carry a spin current[10,11]. Josephson coupling between ICPs has been realized in 2D TMD superconductor van dew Waals (vdW) heterostructures, such as $NbSe_2$ heterostructures with a stacked interface[12] or using a graphene layer as a weak link[13], and suspended $MoS_2$ bilayers with electrical gating[14]. However, the spin-dependent coupling in Josephson characteristics originating from ICPs has not been realized in these systems. In this work, we couple the 2D $NbSe_2$ to the magnetic insulartor $Cr_2Ge_2Te_6$, which enables phase modulation of the spin wave functions of ICPs. The atomically sharp vdW interfaces in $NbSe_2/Cr_2Ge_2Te_6/NbSe_2$ allows momentum-conserving tunneling and leads to a doubly degenerate non-trivial JJ phase.

## Results

**Device characterization**. Van der Waals heterostructures are ideal platforms for creating atomically thin Josephson coupled systems. Here, we use few atomic layers of the vdW magnetic insulator (MI) $Cr_2Ge_2Te_6$[15] as the magnetic barrier to observe a novel Josephson coupling. $NbSe_2/Cr_2Ge_2Te_6/NbSe_2$ heterostructures (Fig. 1a, b)[16] were assembled with a modified dry-transfer technique (see Methods for device fabrication). Our heterostructures clearly display Josephson coupling across $Cr_2Ge_2Te_6$ barriers ranging from monolayer (ML) to 6-ML. Figure 1c–e shows the current density ($J$) versus voltage ($V$) characteristic across the JJs with 1-, 2-, and 6-ML MI barriers. For all devices, we find a clear Josephson supercurrent regime at low bias current, which turns into normal conduction at high bias current. The $J$–$V$ characteristic is hysteric, indicating a switching current density $J_C$ (transition from superconducting to normal state) larger than the retrapping current density $J_R$ (transition from normal to superconducting state). $J_C$ becomes considerably larger than $J_R$ for thinner junctions, which is expected since the junction capacitance is larger for smaller the thickness of F-layer $d_F$. For $J > J_C$, we obtain the normal state resistance $R_N = (1/A)dV/dJ$, where $A$ is the effective area of the junction. Figure 1f shows the comparison of $R_NA$ obtained from devices with three different $d_F$. We find an exponential increase of $R_NA$, fitted well to $a \exp(d_F/t)$, where the characteristic quasiparticle tunneling length is $t \approx 1.3$ nm and the normalized barrier resistance is $a \approx 0.34$ kΩ μm². Importantly, our junction resistance is much lower than that of a typical non-vdW ferromagnetic barrier such as EuS ($10^7 – 10^9$ Ω μm² for the thickness of 2.5 nm)[17]. This relatively small junction resistance is consistent with the smaller semi-conducting energy gap of $Cr_2Ge_2Te_6$ (~0.4 eV in plane and ~1 eV out-of-plane)[18]. Interestingly, we find that while $R_NA$ increases exponentially with increasing $d_F$, the critical current density $J_C$ decreases more rapidly. Figure 1f shows that the product $V_C = J_CR_NA$ decreases exponentially with increasing $d_F$, following $V_C = V_0 \exp(-d_F/\xi_F)$ with the prefactor $V_0 \approx 0.8$ mV and a characteristic

barrier tunneling length in $Cr_2Ge_2Te_6$ $\xi_F \approx 1.4$ nm. While $V_0$ is comparable to $V_C \sim 0.65$ mV in $NbSe_2/graphene/NbSe_2$ junctions[13], the rapid decrease of $V_C$ with increasing $d_F$ indicates that the JJ coupling becomes weaker with a thicker magnetic barrier, as expected.

**Magnetic Josephson junction**. To demonstrate the effect of ferromagnetism in $Cr_2Ge_2Te_6$, we measure the JJ critical current as a function of applied magnetic field. Figure 2a, b shows the in-plane and out-of-plane magnetic-field-dependent switching current $I_C = J_CA$ of the $NbSe_2/Cr_2Ge_2Te_6(6 ML)/NbSe_2$ JJ. We observe hysteretic behavior of $I_C(H)$ for both field directions. $I_C(H)$ also shows a sudden drop near zero magnetic field. Interestingly, we find that the hysteresis in the magnetic field reaches values of ~ ±1.5 T, much larger than the saturation field (the field required to reach the saturation magnetization) of our $Cr_2Ge_2Te_6$ bulk crystals (Fig. 2c), and that of reported values in bulk crystals and thin flakes[15,19,20]. At high bias, which far exceeds the critical current, the voltage across the junction does not show notable hysteresis (Supplementatry Note 1, Supplementary Fig. 2). Furthermore, the magnetization of our bulk $Cr_2Ge_2Te_6$ shows neither a notable hysteresis nor a strong magnetic anisotropy, consistent with earlier reports[19]. The larger hysteresis field compared to the saturation magnetic field of $Cr_2Ge_2Te_6$ and the strong anisotropy observed in $I_C(H)$ thus cannot be simply attributed to the magnetization of $Cr_2Ge_2Te_6$ alone. Rather, the large hysteresis loop for $I_C(H)$ can be related to the microscopic magnetic domain structure of $Cr_2Ge_2Te_6$. Using Lorentz transmission electron microscopy (see Methods for details), we find that thin $Cr_2Ge_2Te_6$ flakes develop two different magnetic domain structures: stripe-like (Fig. 2d) and bubble-like (Fig. 2e). The characteristic domain size is ~100 nm, consistent with previously reported multiple domain structures in thicker $Cr_2Ge_2Te_6$ flakes, where the stripe-phase is more stable than the metastable bubble phase[21]. We conclude that the interplay between the magnetic domains of $Cr_2Ge_2Te_6$ and the field-dependent Abrikosov vortex lattice in $NbSe_2$ can induce a transition between magnetic states and explain the experimental observations, including the sudden drop in $I_C(H)$ near zero magnetic field. This critical current drop can be attributed to the differences in the system energy for the two different vortex states, which is interacting underlying magnetic domains (see Supplementary Fig. 1 and Supplementary Note 1 for detail).

In a Josephson coupling, as a phase difference $\varphi$ develops between two superconductors, a DC Josephson supercurrent $I_S = I_C \sin(\varphi)$ flows through the junction. At equilibrium, the vanishing supercurrent at the minimum energy imposes the condition that $\varphi$ can only be 0 or $\pi$. For conventional superconductors with spin-singlet pairing, the spatially symmetric Cooper pair wavefunction enforces $\varphi = 0$ as the ground state. When the superconducting (S) electrodes are separated by a ferromagnetic barrier (F), Cooper pairs can acquire an additional phase when tunneling through the magnetic barrier, yielding a spatial oscillation of the superconducting order parameter in the barrier[22–24]. Tuning the thickness of the F-layer, $d_F$, can reverse the sign of the superconducting order parameter across the barrier owing to an exchange-energy driven phase shift[23,25–27], resulting in a π-phase JJ.

**SQUID and $\phi$ phase**. To probe possibly anomalous Josephson phase, we have realized SQUIDs consisting of one MI JJ ($NbSe_2/Cr_2Ge_2Te_6/NbSe_2$) and one reference JJ ($NbSe_2/NbSe_2$). After the assembly, we create the device by etching away the unnecessary areas (Fig. 3a, note the edges of the $NbSe_2$ flakes were aligned parallel to each other (see Methods for details)). The wider MI JJ allows us to balance the critical currents for each JJ for a maximal

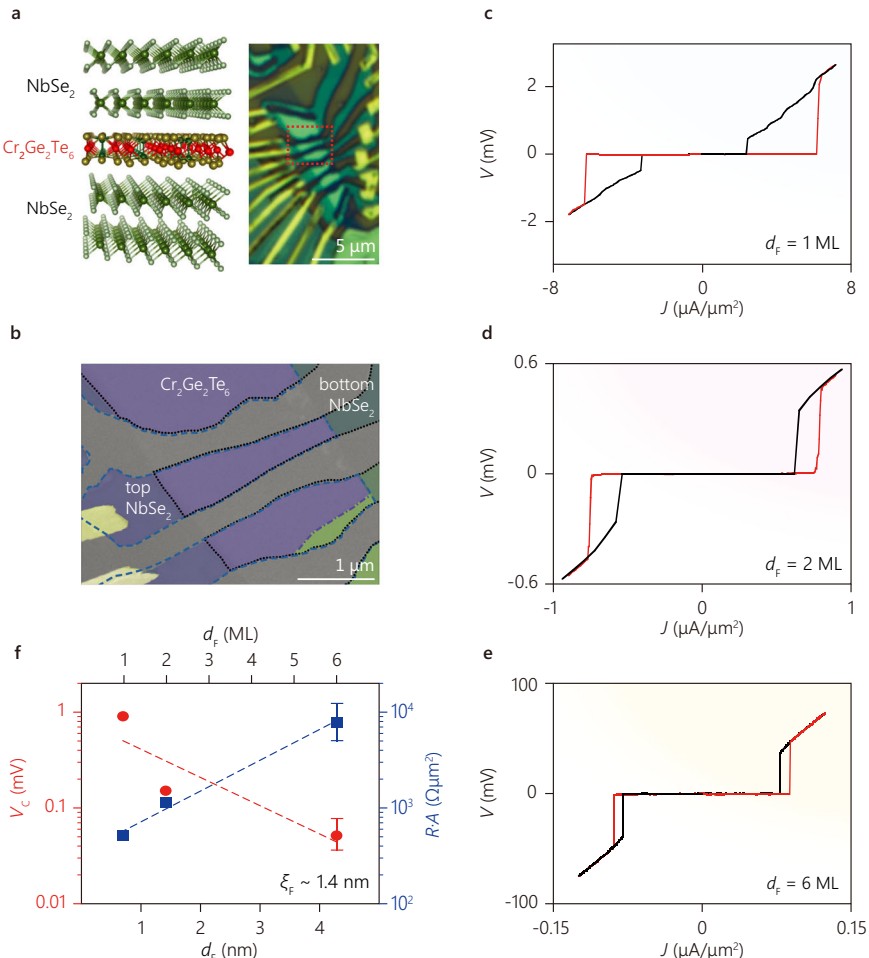

**Fig. 1 Josephson coupling of the NbSe$_2$/Cr$_2$Ge$_2$Te$_6$/NbSe$_2$ van der Waals junction. a, b** Schematic (left panel of (**a**))[16], optical micrograph (right panel of (**a**)), and false color scanning electron microscope (SEM) image (**b**) of NbSe$_2$/Cr$_2$Ge$_2$Te$_6$/NbSe$_2$ Josephson junctions (JJs). In the schematic, top and bottom pairs of layers represent NbSe$_2$ and the middle layer represents Cr$_2$Ge$_2$Te$_6$. In the optical micrograph, the gold contact leads appear as yellow lines and the JJ appears in green. The red dotted line indicates the area of the SEM image. In the SEM image, the areas of the top and bottom NbSe$_2$ flakes are indicated by the blue dashed line and the black dotted line, respectively. The Cr$_2$Ge$_2$Te$_6$ and NbSe$_2$ layers are indicated by purple and green. The dash-dot line indicates the boundary of Cr$_2$Ge$_2$Te$_6$. **c–e** V–J characteristics of the JJs for the Cr$_2$Ge$_2$Te$_6$ barrier thicknesses of 1, 2, and 6 ML. The junction area $A$ is ~0.9 μm$^2$, ~4 μm$^2$, and ~1.4 μm$^2$, respectively. The red lines depict the transition from the superconducting state to the normal state whereas the black ones depict the transition from the normal state to the superconducting state. **f** The characteristic voltage (switching current normal resistance product, red dot) and resistance-area product (blue rectangle) of the junction with different barrier thickness. The thickness of the barrier is shown in units of both nanometers and the number of layers (ML). Fitted result is shown in dashed lines and error bars indicate standard error.

SQUID critical current $I^C_{SQUID}(\Phi)$ as a function of magnetic flux $\Phi$ threaded through the SQUID loop. The critical current measured in the SQUID (Fig. 3b) exhibits oscillations with the periodicity $\Phi_0 = h/2e$. However, we observe an irregular SQUID response in the field range between −1.2 and −2.2 mT, with a telegraph-like signal oscillating between two metastable critical current branches (Fig. 3b, c). This bi-stability is possibly related to the sudden change in critical current seen in Fig. 2a caused by the change of magnetic structure in the junction. This bistable switching state is an indirect indication of a doubly degenerate ground state of the system (see Supplementary Note 2 for details). Nevertheless, the regular oscillation around zero magnetic field allows us to extract the phase of the MI JJ (NbSe$_2$/Cr$_2$Ge$_2$Te$_6$/NbSe$_2$). In a previous study of SQUIDs with ferromagnetic metallic spin valves[28], a controllable switching between 0- and π- Josephson junctions has been demonstrated. A SQUID that combines 0/0 or π/π JJs shows a maximal $I^C_{SQUID}(0)$ (defined as a 0-phase JJ), whereas a SQUID combining 0/π JJs shows a minimal $I^C_{SQUID}(0)$ (defined as a π-phase JJ).

For our SQUID with the MI JJ, we use two schemes to measure the two different switching currents (Fig. 3d): a switching current $I_{C−}$ obtained by sweeping from large negative bias to positive bias and another one, $I_{C0}$, obtained by sweeping from large positive bias to zero and then back to positive bias. Generally, we find $I_{C−} > I_{C0}$. More importantly, the phases of their oscillations are different, as shown in Fig. 3e. To obtain the absolute phase of $I^C_{SQUID}(\Phi)$, we have carefully calibrated our electromagnet for zero magnetic field using several on-chip Al SQUIDs with different sizes (see Supplementary Fig. 7 for details). Strikingly, we find that none of the switching schemes provide 0 or π phase but $\phi_{C−} = 259°$ and $\phi_{C0} = 59°$ as shown in Fig. 3e.

The presence of these two nontrivial phases (i.e., not simple multiples of 180°) is reminiscent of two switching current states in a metallic ferromagnetic (F) $\phi$-JJ[29,30]. It is reported that an arbitrary $\phi$-phase between 0 and π can be realized by engineering the combination of 0- and π-JJs[29]. One example is a long channel metallic F-JJ (typically 100 μm)[30], where the doubly degenerate ground states are realized by laterally connecting 0-junction and

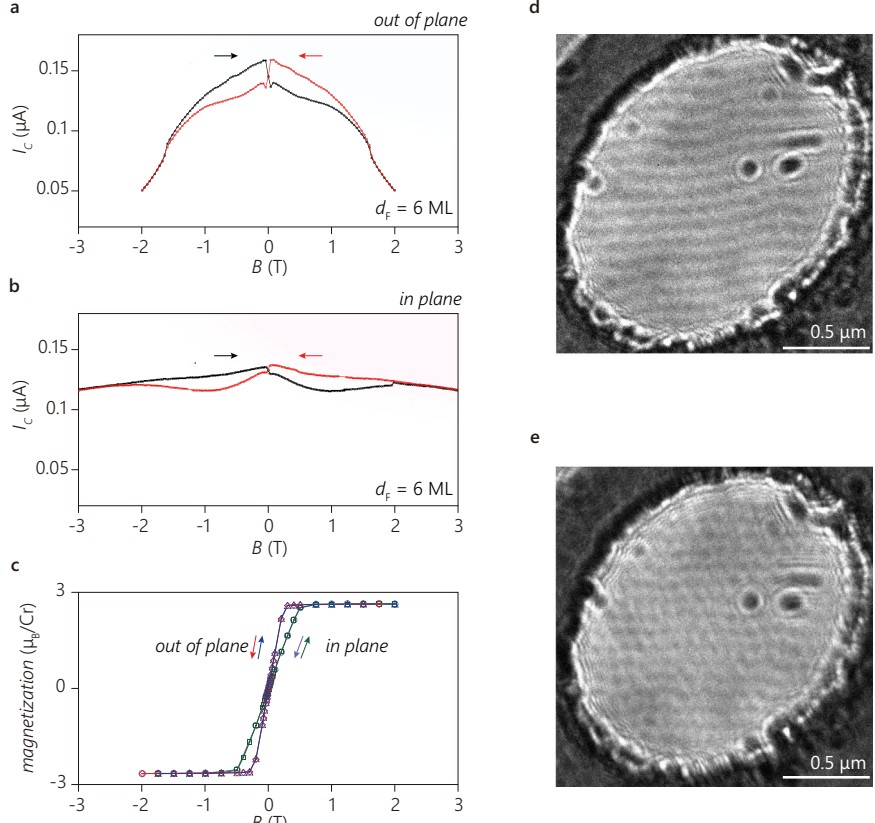

**Fig. 2 The magnetic characteristics of the Josephson Junction and Cr₂Ge₂Te₆. a**, **b** The switching current of the NbSe₂/Cr₂Ge₂Te₆/NbSe₂ junction with a 6 ML Cr₂Ge₂Te₆ barrier at the temperature of 0.3 K as a function of the applied magnetic field (**a**) perpendicular (**b**) and parallel to the 2D layer. The red lines indicate the magnetic field scan from positive polarity to negative polarity and the black ones from negative polarity to positive polarity. **c** The bulk magnetization of a Cr₂Ge₂Te₆ crystal measured by a SQUID magnetometer at 4 K. The red diamonds and blue triangles indicate the data with perpendicular field, and the purple circles and green triangles indicate the data with in-plane field. The arrows indicate the sweep direction for (**a**–**c**). **d** The magnetic domain structure of a Cr₂Ge₂Te₆ flake measured by Lorentz microscopy without an applied magnetic field. The thickness of the flake is 18 nm. The sample was tilted by 45° from the incident electron beam direction to image an out-of-plane magnetization pattern at the temperature of 25 K. The black and white contrast in the central oval-shape area indicates magnetic domain wall contrast caused by the opposite magnetization. The oval-shape is the area where Cr₂Ge₂Te₆ is suspended over a hole in the SiN membrane support. **e** Lorentz micrograph of the same sample with (**d**) but taken with a different thermal cycle. Unlike the simple stripe domains shown in (**d**), bubble-like domains appear to break up the stripe-like domains. The scale bars correspond to 0.5 μm.

π-junction. Here, the π-JJ requires the thickness of the F-layer to be comparable to the wavelength of the order-parameter oscillation, implying $d_F \sim 10$ nm, set by the exchange energy[24]. In addition, the widths of the 0-JJ and π-JJ perpendicular to the supercurrent flow direction is restricted to be much longer than the Josephson length, $\lambda_J = \sqrt{\Phi_0/2\pi\mu_0 J_C\lambda_C}$, where $\lambda_C$ is the magnetic penetration length, because the formation of a $\phi$-FJJ in the 0-π JJ arrays needs a Josephson vortex pinned at the 0-π junction. Such values of $d_F$ and $\lambda_J$ are incompatible with our atomically thin Cr₂Ge₂Te₆ based JJ. Specifically, our Cr₂Ge₂Te₆ barrier is an atomically thin insulator ($d_F = 1$ nm), which is too thin to exhibit spatial order-parameter oscillations. Furthermore, the lateral size of our JJ, $L < 5$ μm, is much smaller than $\lambda_J$ ~10 μm, estimated using our experimentally obtained $J_C$ and $\lambda_C \approx 0.1$ μm for NbSe₂ reported previously[31]. Therefore, the observed $\phi$-JJ formation in our atomically thin MI-JJ, manifested by the appearance of doubly degenerate nontrivial phase shifts, requires an alternative mechanism.

**Interplay between Ising superconductivity and ferromagnetism.** The single-crystallinity of our vdW heterostructure combined with the strong SOC in the quasi-2D superconductor (S)

constituent provides two new characteristics for Josephson coupling that are absent in conventional metallic F-JJs. First, in contrast to the F-JJs constructed by sputtered heterostructures[32], momentum-conserving tunneling in crystalline vdW heterostructures is allowed between the closely aligned Fermi surfaces of two S-layers in vdW JJ, as the top and bottom S layers in our JJ are aligned along the same crystallographic axis (< 2°–5° misalignment; see "Methods"). Second, the strong SOC in NbSe₂ fixes the spin quantization axis of the Cooper pairs[6,33] normal to the substrate, denoted by ↑ and ↓. In NbSe₂, due to weak interlayer tunneling and strong inversion symmetry breaking within each layer, two spin components remain localized predominantly in even or odd layers with a sizable spin splitting $\Delta_{SOC} \simeq 100$ meV[7]. This spin-layer locking results in unconventional Ising Cooper pairing ($K\uparrow$–$K'\downarrow$ or $K\downarrow$–$K'\uparrow$) inside each layer where $K$ and $K'$ denote the electronic band near $K$ and $K'$ points.

The Josephson phase between ICPs on the surfaces of NbSe₂ across the MI barrier can be sensitively modified by the magnetization direction. For out-of-plane magnetization, the spin of the ICPs is aligned parallel or antiparallel with the spin of the MI. Similar to a previous theoretical study of JJs with magnetic impurities[34], the wave function of ICPs tunneling across the ferromagnetic junction can acquire an additional minus sign

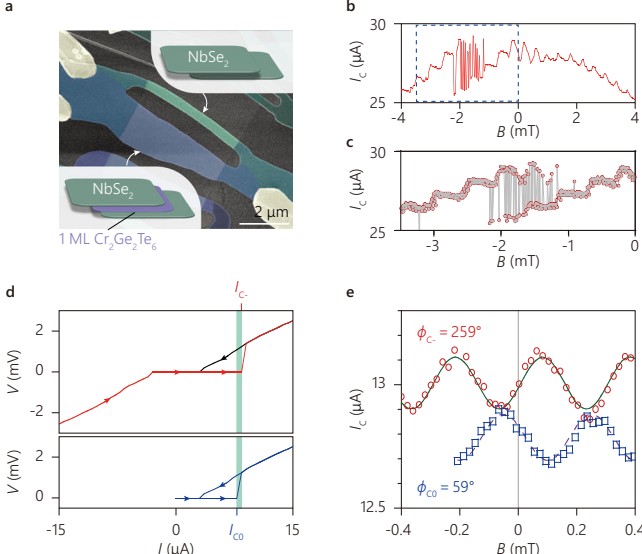

**Fig. 3 Josephson phase measurement of the vdW SQUID devices with ultrathin magnetic insulator junction. a** False-color SEM image of the NbSe$_2$ SQUID device with and without a 1 ML Cr$_2$Ge$_2$Te$_6$ barrier (Cr$_2$Ge$_2$Te$_6$-SQUID). The inset shows a schematic of cross-section of the junctions indicated by arrows. **b** The SQUID oscillates with telegram-like critical current between the fields of −1.1 and −2.2 mT, which implies the presence of two metastable state. The loop area (≈4.5 μm$^2$, including the area for screening ≈2 μm$^2$) agrees with the area calculated from the oscillation period ~4-6 μm$^2$. **c** The detailed view of the switching region of bistable SQUID critical current oscillations marked by dashed box in (**b**). **d** The SQUID device is similar to the one shown in (**b**) but with a smaller critical current. The sweep-dependent critical currents for Cr$_2$Ge$_2$Te$_6$-SQUID measured at $T = 1.1$ K. The red line in the top panel depicts the sweep from negative bias (current) to positive bias, with switching current $I_{C−}$. The black line depicts the sweep from positive bias to zero bias. The blue line in the bottom panel depicts the sweep from positive bias to zero bias followed by a sweep from zero bias to positive bias, whose switching current $I_{C0}$ is lower than $I_{C−}$. The green area indicates the difference of the switching currents. **e** The Cr$_2$Ge$_2$Te$_6$-SQUID oscillation. The loop area (≈7 μm$^2$, including the area for screening ≈4.5 μm$^2$) agrees with the area calculated from the oscillation period ~7 μm$^2$. The field is calibrated by three different sizes of aluminum SQUIDs located 0.1 mm away from the Cr$_2$Ge$_2$Te$_6$-SQUID. This allows for precise calibration of zero field and thus measurement of the phase of the SQUID oscillation. Red circles and blue rectangles represent $I_{C−}$ and $I_{C0}$ measured at $T = 0.9$ K. Each branch has a different phase, indicated by a fit to $I_{C−}$ and $I_{C0}$ with the sine curves shown in green and purple.

with respect to the nonmagnetic junction for sufficiently strong magnetic scattering (Fig. 4a, see "Methods" for details). Importantly, this sign flip of the Josephson coupled ICPs sets the phase of the JJ ground state to be $\varphi = \pi$. As the magnetization of the MI-layer is tilted away from the tunneling direction, the spin of the ICPs can flip during the tunneling process (see "Methods" and Supplementary Note 4 for details). As a result, the ground state of the Josephson junction is at $\varphi = 0$ (Fig. 4b). Unlike metallic F-JJs, our heterostructure allows for both 0- and π-JJs by adjusting the direction of the magnetization in the MI.

A parallel arrangement of 0- and π-JJ can lead to a degenerate $\phi$-JJ. Our MI-JJ offers such lateral arrays, created by magnetic domain structures in the MI, similar to what is shown in Fig. 2d,e. Here, the domains with out-of-plane magnetization separated by boundaries with tilted spins could lead to a coexistence of 0 and π junction segments. Using a simple model based on a short junction with finite transparency $D$ and a fraction of the plane $\lambda$

that favors a π junction, we can estimate the Josephson energy $E_J$ of the junction, which provides two $\phi$ values for the degenerate ground states (see Methods section for more details). As an example, Fig. 4d shows $E_J(\phi)$ computed using this model with $\lambda = 0.53$ and $D = 0.75$, resulting in the appearance of two ground states at different nontrivial Josephson phases $\phi_1 \simeq 100°$ and $\phi_2 \simeq 260°$. These two states can be obtained by sweeping back from positive or negative bias as the switching currents can be simply controlled by choosing two metastable states in the bi-stable potential. The telegram-like signal in Fig. 3b is found in the negative magnetic field region. A careful examination of this two state switching behavior implies that two meta-stable SQUID oscillations with different phases involve (Fig. 3c), suggesting that the bistability can also be controlled by applied magnetic fields. We also note that the experimental value of $\phi_1$ deviates substantially from the theoretical value obtained above. This can be attributed to the experimental anisotropy and inhomogeneity of the devices as observed in the in-plane field Fraunhofer pattern (see Supplementary Note 3), which is not considered in our theoretical analysis above.

## Discussion

Experimentally, the presence of two minimal phase angles in $E_J(\phi)$ can be directly revealed by measuring the JJ switching current distributions. This distribution is sensitively determined by the escape rate $\tau^{-1}$ from a tilted washboard potential that is created by biasing $E_J(\phi)$ with current $I$ (insets of Fig. 4f). Figure 4e shows the switching current distribution measured using the two above-mentioned sweep schemes, as a function of current $I(t)$ increasing monotonically with time $t$. The switching current distribution shows not only different values of the critical current but also a much wider distribution for the $\phi_{c0}$ state than for the $\phi_{c−}$ state (the bottom panel of Fig. 4e). Figure 4f shows the escape rate for both ground states, which is calculated using the normalized distribution function $P(I_C)$ and the Fulton and Dunkleberger formula $\tau = (1 - \int_0^I P(u)du)/[P(I)\frac{dI}{dt}]$[35]. Generally, we find the escape rate for the $\phi_{c0}$ state to be larger than for the $\phi_{C−}$ state, suggesting the $\phi_{C−}$ state is more stable than the $\phi_{C0}$ state under a bias current. Assuming the escape process is dominated by thermal activation, it gives $\tau^{-1} \sim e^{-\Delta/kT}$, where $\Delta$ is the barrier height in the tilted washboard potential and $k$ is the Boltzmann constant. Since the experimentally estimated $\tau^{-1}$ is smaller for $\phi_{C−}$ than that for $\phi_{C0}$, we infer that $\Delta_1 < \Delta_2$ (Fig. 4d), where $\Delta_1$ is responsible for the switching of $\phi_{C0}$ and $\Delta_2$ for the switching of $\phi_{C→}$, in agreement with the model presented in Fig. 4d. For an applied bias current $I$ smaller than the switching current of the $\phi_{C−}$ state, re-trapping is allowed (inset of Fig. 4f), which consistently explains the slower increase of $\tau^{-1}$ of the $\phi_{C0}$ state before the switching of the $\phi_{C−}$ state at higher current.

To date, a metallic ferromagnetic barrier requires $d_F \geq 5$ nm and a macroscopic lateral junction size, which is not in general suitable for use in dissipationless and compact quantum device components. JJs using magnetic semiconducting GdN barriers in the spin filter device geometry[32] exhibit an unconventional second harmonic current-phase relation and switching characteristics[36,37]. In contrast, we have demonstrated a Josephson phase engineering in dissipationless magnetic JJs. $\phi$-phase JJs can serve as useful components for various superconducting quantum electronic devices, such as phase batteries that can be used to bias both classical and quantum circuits, superconducting-magnet hybrid memories and JJ-based quantum ratchets[28,38–40]. The spin sensitivity of an Ising Josephson junction together with atomically thin magnetic tunneling barriers provides a route to the fabrication of novel superconducting and spintronic devices.

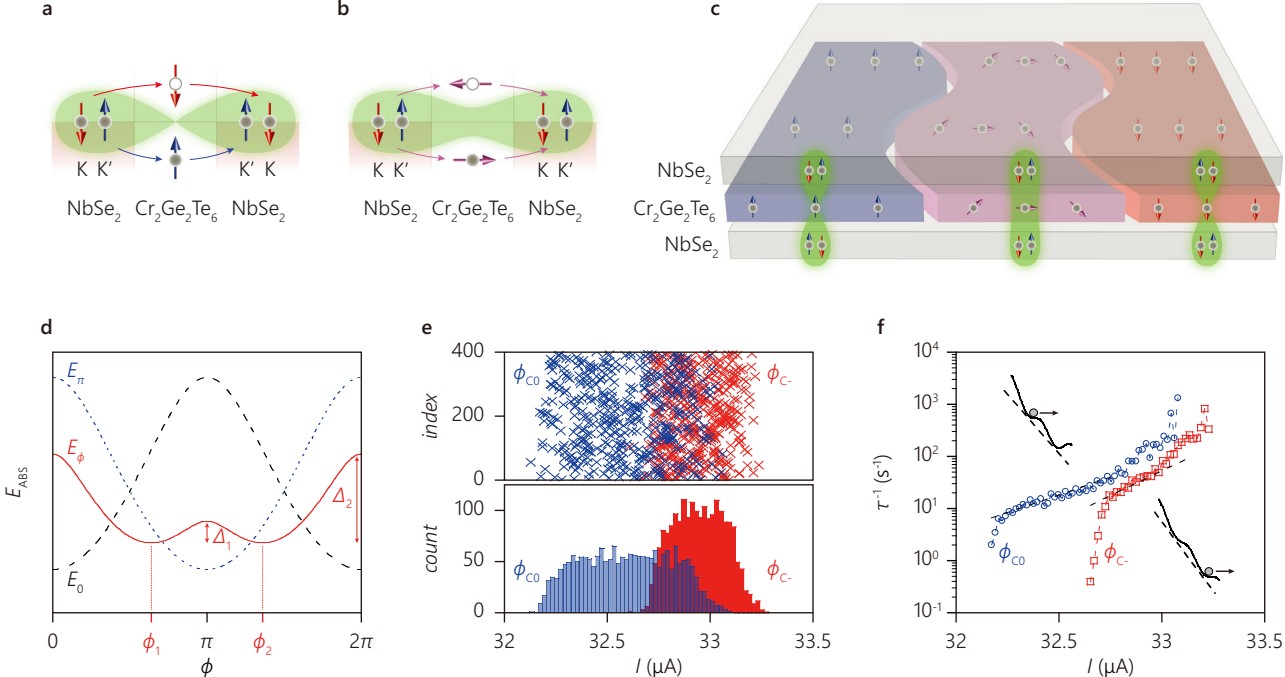

**Fig. 4 Ising-Cooper-pair coupling through the all-crystalline vdW magnetic Josephson junction. a–c** Illustration of the Ising-Cooper-pair coupling through the magnetic tunneling junction for perpendicular magnetization (**a**), for in-plane magnetization (**b**), and for magnetic structure (**c**). For perpendicular magnetization, Ising Cooper pairs can tunnel via the spin-dependent energy levels without spin flip, forming a π-phase junction. For in-plane magnetization, tunneling occurs with a spin flip, which requires zero phase. With the magnetic domain structure of $Cr_2Ge_2Te_6$, the Josephson junction consists of segments of π- and 0-phases. **d** Washboard potential for the $\phi$ junction (indicated by solid red line), zero junction (dashed black line), and π junction (dotted blue line). Nontrivial Josephson phase $\phi_1$ and $\phi_2$ appear as the minimum energy for $\phi$ state (the red line is calculated using $\lambda = 0.53$ and $D = 0.75$). **e, f** switching current distribution for the $\phi$ state at $T = 1.1$ K. Different switching current is repeatedly acquired with different sweeping, as depicted in Fig. 3d. The lower and higher critical currents are depicted in blue and red. **e** Switching currents with different sweeping events (top panel) and histogram of switching currents in 0.02 μA bins (bottom panel). **f** Escape rate $\tau^{-1}$ as a function of the bias current that has a different slope (corresponding to an energy higher than the environment temperature) reflected by the different metastable states of the tilted washboard potential shown in (**d**), and the potential barrier for each escaping event is schematically illustrated in the insets. The dashed lines are guide to eyes.

## Methods

**Crystal synthesis**. $NbSe_2$ crystals were grown from pre-cleaned elemental starting materials in an evacuated quartz glass tube, in a 700 to 650 °C temperature gradient, by iodine vapor transport. $Cr_2Ge_2Te_6$ was grown out of a ternary melt that was rich in the Ge–Te eutectic. High purity elements we placed into fritted alumina crucibles[41] in a ratio of $Cr_5Ge_{17}Te_{78}$, sealed in an amorphous silica ampoule under roughly 1/4 atmosphere of high purity Ar. The ampoule was heated over 5 h to 900 °C, held at 900 °C for an additional 5 h, and then cooled to 500 °C over 99 h. At 500 °C, the excess liquid was separated from the $Cr_2Ge_2Te_6$ crystals with the aid of a centrifuge[41]. The single crystals grew as plates with basal plane dimensions of up to a cm and had mirrored surfaces perpendicular to the hexagonal c-axis (inset to Supplementary Fig. 6). Low field magnetization on a bulk sample is shown in Supplementary Fig. 6 and is consistent with a ferromagnetic transition near 65 K.

**Device fabrication**. $NbSe_2$ and $Cr_2Ge_2Te_6$ crystals of the desired thickness were mechanically exfoliated onto a p-doped silicon chip terminated with 285 nm $SiO_2$. The following fabrication procedure was employed unless explicitly noted otherwise. Exfoliated crystals were identified by optical contrast (some of them were separately characterized with atomic force microscope) in an argon-filled glove box. The thickness of $NbSe_2$ flakes ranges from 8 to 16 nm (on average 12 nm) except for the top flake of the 2-ML junction (100 nm) and the $NbSe_2/NbSe_2$ junction in Supplementary Note 2. The $NbSe_2/Cr_2Ge_2Te_6/NbSe_2$ heterostructure was prepared by polymer-based dry transfer technique, inside of the Ar glove box, with maximum process temperature of typically between 60 and 80 °C so that degradation of the flakes is prevented[42]. The surface of the stack was examined by atomic force microscopy and/or scanning electron microscopy to identify the clean and atomically flat parts of the junction. Unnecessary parts were removed by reactive ion etching with fluorine gas using an electron (e)-beam (Elionix ELS-F125) patterned mask. For the SQUID device with field-calibration sensors, a double-layer resist was patterned by e-beam lithography followed by oblique deposition of aluminum. To form $Al/Al_2O_3/Al$ junctions, an in-situ oxidation process (1 mTorr, 10 min) was utilized between the e-beam evaporation of Aluminum. After the lift-off of the e-beam pattern, Ti/Au contacts were patterned by e-beam lithography using a polymethyl methacrylate mask followed by e-beam

evaporation to contact both $NbSe_2$ and Al. Before this evaporation, the surface was in-situ cleaned by ion-milling.

**Preparation of the states for different switching current**. For the measurement in Fig. 3e, the switching current branch was prepared as follows: first, the device was measured by specific bias sweeps (from positive bias to negative bias, then returning to zero bias) at a magnetic field of 8.2 mT. Then we swept the magnetic field within the range between −0.4 and 0.4 mT to determine the switching current. The other switching current branch was characterized in the same manner, but the initial state was prepared by sweeping from positive bias to zero bias (at the same field of 8.2 mT).

**Lorentz transmission electron microscopy**. $Cr_2Ge_2Te_6$ flakes were obtained by mechanical exfoliation and transferred onto 50-nm-thick SiN membranes with holes supported by an Si frame in an Argon-filled glove box, by a similar procedure to the device fabrication. The samples were mounted on the liquid-Helium-cooling holder (ULTDT, Gatan) and inserted to the 300-kV field-emission TEM (HF-3300S, Hitachi High-Tech) specially designed for eliminating the magnetic field in the sample area. The Lorentz micrographs were taken using a defocusing condition in the electron optical system and the sample tilting condition (45°–30° tilt measured from the normal position for the optical axis)[43]. The thin flake (approximately 18-nm-thick) over the hole showed the disappearance of the stripe pattern near the holder temperature of 62 K, consistent with Curie temperature of $Cr_2Ge_2Te_6$.

**Theoretical model**. The spectrum of monolayer $NbSe_2$ has Fermi pockets around the $\Gamma$ point and around the $K$ and $K'$ points, and the bands around the $K$ and $K'$ points have sizable spin splitting $\Delta_{SOC} \simeq 100$ meV due to inversion-symmetry breaking spin–orbit interaction, which results in Ising superconductivity[7]. In contrast, bulk $NbSe_2$ has an inversion center between the layers, such that the SOC is opposite in even and odd layers destroying the spin-momentum locking. Because of weak interlayer hopping, however, bulk $NbSe_2$ can be thought of a stack of weakly coupled Ising superconductors with opposite spin polarization in even and

odd layers. Tunneling in a Josephson junction predominantly originates from the layer adjacent to the junction and as a result the supercurrent will be carried by ICPs. There can be non-vanishing contribution from the $\Gamma$ point in the Josephson coupling, which modifies the relative amplitudes of $\pi$- and 0-couplings of the ICP (see Supplementary Note 4).

In our phenomenological model, we neglect any coupling between the layers. An effective low-energy Hamiltonian for the $K$ and $K'$ valleys in a single layer can be written in Bogoliubov-de Gennes form in the Nambu-spinor basis $\psi = (\psi_\uparrow, \psi_\downarrow, \psi_\uparrow^\dagger, -\psi_\uparrow^\dagger)$ as

$$H_{SC} = \nu[p_\parallel - (p_F + p_{SOC}\sigma_z\lambda_z)]\tau_z + \Delta[\cos(\phi/2)\tau_x - \sin(\phi/2)\tau_y] \quad (1)$$

where $\sigma$, $\tau$, and $\lambda$ are Pauli matrices acting in spin, particle-hole and valley space, respectively. The SOC is opposite in the two valleys thus preserving time-reversal symmetry. The superconducting phase $\phi/2$ has opposite signs in the two leads, such that $\phi$ is the phase difference across the junction. For concreteness, we here assume that the SOC has the same sign both sides of the junctions. In the case of opposite signs, a similar argument for a $\phi$ junction can be made. We moreover assume the pairing strength $\Delta$ to be small compared to the spin splitting $2\nu\, p_{SOC}$ and hence the Cooper pairs consist of two electrons with opposite spins aligned with the $z$ direction. The magnetic layer is approximated by a single insulating band for each spin, whose energy bands are flat in two-dimensional momentum space. In the Nambu spinor basis $(d_\uparrow, d_\downarrow, d_\downarrow^\dagger, -d_\uparrow^\dagger)$, the Hamiltonian reads

$$H_{MI} = V\tau_z + J\vec{\sigma} \cdot \mathbf{n}, \quad (2)$$

where $V$ and $J$ denote the potential and exchange energy and $\mathbf{n}$ is a unit vector describing the direction of the magnetization. An extension to more complicated band structures is possible but will not qualitatively change our conclusions. The superconductors and the magnet are coupled by the hopping Hamiltonian

$$H_T = \sum_\sigma (t\psi_{L,\sigma}^\dagger d_\sigma + t\psi_{R,\sigma}^\dagger d_\sigma + h.c.) \quad (3)$$

where $t$ is positive. We now calculate the spectrum of Andreev bound states in the junction. For off-resonant tunneling, $t \ll V, J$, we can obtain an effective hopping between the left and right superconductor from second-order perturbation theory

$$H_{T.eff} = \tilde{t}\psi_{L,\sigma}^\dagger \psi_{R,\sigma} + h.c., \quad (4)$$

where the effective hopping strength is

$$\tilde{t} = \frac{t^2 V}{V^2 - J^2} - \frac{t^2 J\vec{\sigma} \cdot \mathbf{n}}{V^2 - J^2}. \quad (5)$$

If the junction is nonmagnetic, $J = 0$, we obtain $\tilde{t} = t^2/V$ and the Andreev spectrum simply is that of a narrow Josephson junction in a BCS superconductor.

$$E = \pm\Delta\sqrt{1 - D\sin^2\phi/2}, \quad (6)$$

where the transparency is $D = \pi^2\tilde{t}^2\nu^2/(1 + \pi^2\tilde{t}^2\nu^2)$ with $\nu$ normal density of states in the superconductors. This is a regular Josephson junction whose ground state is at $\phi = 0$. For a purely magnetic junction with $\mathbf{n}$ along the $z$ axis, we instead obtain an effective hopping parameter

$$\tilde{t} = \frac{t^2\sigma_z}{J}. \quad (7)$$

The hopping has a different sign for the two spin components and, hence, a Cooper pair tunneling across the junction acquires an additional minus sign with respect to a nonmagnetic junction. We can show this explicitly by doing a gauge transformation $\psi_{L,\downarrow} \to -\psi_{L,\downarrow}$ while leaving all other fermions invariant. In this new gauge the hopping is nonmagnetic $\tilde{t} \to t^2/J$ and the pairing term in the left superconductor changes sign $\Delta_L = \langle\psi_{L,\uparrow}, \psi_{L,\downarrow}\rangle \to -\Delta_L$ while the remaining terms are unchanged. This shows that we obtain the same spectrum as before but with a $\pi$ phase shift

$$E = \pm\Delta\sqrt{1 - D\sin^2(\phi + \pi)/2}. \quad (8)$$

Hence the ground state of the Josephson junction is at $\phi = \pi$. In fact, the system always forms a $\pi$ junction when $J > V$ as was first noted in ref. [44].

Now we consider a junction with magnetization along the $x$ direction so that scattering in the junction can result in spin flips. Due to the strong SOC, however, the band structure in the superconductor is helical, meaning that at any particular in-plane momentum there is only one spin component at the Fermi level. Thus, if a spin flip occurs in the barrier the other spin component has a large momentum mismatch when entering the superconductor. The latter therefore acts as a hard wall for flipped spins as long as the superconductor-magnet interface is sufficiently clean, such that scattering approximately conserves in-plane momentum. Andreev reflection can therefore only happen after an even number of spin flips in the barrier, which means the supercurrent is an even function of $J$ in this case and all spin dependence drops out. This implies in particular that hopping has the same sign for electrons with different spins and hence the junction always has a ground state at zero.

Now let us assume that the magnet is inhomogeneous and there are regions with magnetization along $z$ and $x$. This means critical current changes sign as a function of the in-plane position. When the length scale of the spatial variations is smaller than the Josephson screening length the critical current is simply the spatial average of the current. As a simple model, we assume a fraction $\lambda$ of the plane favors a $\pi$ junction described by Eq. (8). The remaining fraction $(1 - \lambda)$ is instead described by Eq. (6). Note that the latter also includes a conventional Josephson current due to electron near the $\Gamma$ point. In Fig. 4d we plot the spectrum of the Josephson junction when the transparency is $D = 0.75$ in both regions and $\lambda = 0.53$. See the Supplementary Note 4 for the microscopic description of the theoretical model.

## Data availability

The datasets generated during and/or analyzed in the current study are available from the corresponding author upon reasonable request.

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

## Acknowledgements
We thank K.F. Mak, J. Shen, and Y. Otani for a fruitful discussion. The major part of the experiment performed by H.I. was supported by ARO (W911NF-17-1-0574) and measurement performed by K.F.H. was supported by NSF (QII-TAQS MPS 1936263). The sample fabrication was supported by DOE QPress (DE-SC0019300). P.K. acknowledges support from the DoD Vannevar Bush Faculty Fellowship N00014-18-1-2877. H.I. acknowledges JSPS Overseas Research Fellowship and the Nakajima Foundation for support. L.T.N. and R.J.C. acknowledge the US Department of Energy, Division of Basic Energy Sciences, grant DE-FG02 98ER45706 for supporting the growth of the NbSe₂ crystals. Work done at Ames Lab (P.C.C. and N.H.J.) was supported by the U.S. Department of Energy, Office of Basic Energy Science, Division of Materials Sciences and Engineering. Ames Laboratory is operated for the U.S. Department of Energy by Iowa State University under Contract No. DE-AC02-07CH11358. N.H.J. was supported by the Gordon and Betty Moore Foundation's EPiQS Initiative through Grant GBMF4411.

## Author contributions
H.I. fabricated the sample and analyzed the data. H.I. and K.-F.H. performed the measurements. K.H. and D.S. performed TEM experiments. H.I. and P.K. conceived the experiment. F.P. developed the theoretical description. Y.J.S. provided the polymer and optimized transfer process. N.H.J. and P.C.C. grew and characterized single crystals of Cr₂Ge₂Te₆. L.T.N. and R.J.C. grew the NbSe₂ crystals. Ö.G. contributed to the interpretation of the results. H.I. and P.K. wrote the paper with input from all other authors.

## Competing interests
The authors declare no competing interests.
