## [Peer Review File · Nature Communications]

REVIEWER COMMENTS

Reviewer #1 (Remarks to the Author):

In this manuscript, Izuchi et al. demonstrated Josephson junction composed of superconductor/ferromagnetic insulator/superconductor (SFS). In this junction, they utilized layered material such as NbSe₂ and CrGeTe to construct van der Waals SFS junction. It is also noticeable the NbSe₂ is having rather unique properties as a superconductor since it is highly anisotropic (Cooper pair spins are aligned to out-of-plane) and noncentrosymmetric in case of few-layer NbSe₂. This system under investigation is novel and could be interesting to explore novel physics originated from superconductor/ferromagnet interface. As the author claimed, use of van der Waals interface can significantly reduce the imperfections in the interface compared to the junction fabricated by a metal deposition technique. The results of this study could be interesting to the community of both superconductor and two-dimensional material sciences. However, I feel that some of the explanations and interpretation of the results are not convincing enough and need to be improved. I would recommend the author to add an explanation for the following points.

1) What is the thickness of the NbSe₂?

2) The hysteresis observed in the junction Josephson critical current presented in Fig. 2(a,b) looked interesting and may capture something exotic phenomena. However, I am not sure about their interpretation presented in supplementary information.

(a) First, the magnetism can change with its size. Therefore, I would imagine that their thin Cr₂Ge₂Te₆ can have different saturation fields compared with its bulk form. It may even have hysteresis in thin film. Although authors discuss that hysteresis they observed appears to have a different magnetic field range with the saturation field of bulk Cr₂Ge₂Te₆, it may not rule out the possible contribution of the magnetism of CrGeTe material. Please add some explanation or some reference that studies the size dependence of magnetism on CrGeTe(if available).

(b) Secondly, I do not understand why the explanation in Fig. S1 can cause the change of critical current in the SFS junction. Author discusses the change of magnetic domain pattern in CrGeTe due in relation to the vortex formation in NbSe₂. It would be great if the author can explain in more specifically how their explanation increases or decreases the critical current.

(c)The explanation presented in Fig. S1 does not depend on the junction bias (or junction current) as far as NbSe₂ is superconducting and disappears when NbSe₂ is in a normal state. Does the author have extra data that shows hysteresis disappears when NbSe₂ is in normal states such as applying large current or increasing measurement temperature?

(d)The vortex formation process in Fig. S1 should be different for the application of an in-plane magnetic field. However, the author observed a similar range of hysteresis in Fig. 2(a) and 2(b) accompanying a similar sudden jump at low field. I wonder how they can explain the behavior on Fig. 2(b) with their model.

(3)From their explanation in the method section, I am assuming their NbSe₂ is thick bulk crystal. The multilayer NbSe₂ is having Fermi level at Gamma-point, not only the K-point. Some of the previous study probably suggested that superconductivity possibly appears in both Gamma-point and K-point. Author presented the argument shown below to exclude the contribution from Gamma-point, but I am not sure whether this is really true. Is there any experimental result (could be reference paper) that can justify their assumption that contribution from gamma point is negligible.

“Because of weak interlayer hopping, however, bulk NbSe₂ can be thought of a stack of weakly coupled Ising superconductors with opposite spin polarization in even and odd layers. Tunneling in a Josephson junction predominantly originates from the layer adjacent to the junction and as a result the supercurrent will be carried by ICPs.”

(4)I believe that the “doubly degenerate ground state” they called appeared Fig. 2(a) and 2(b) as a

sudden jump at a low magnetic field, thus can be prepared by magnetic field. In the later section (Fig. 3), they prepared these two states by controlling current. It makes it a bit difficult how to interpolate these states prepared by two different sequences are the same. It should be more simpler if the author prepared the initial state in Fig. 3 with a magnetic field, but they must have some reason not to do that. I recommend the author to add more explanation to explain the relationship between the two experiments. I also wonder why the current sweep can change the initial state.

Overall, results seem to be new and possibly could promote further discussion in the research community. I would think this can be a good manuscript if the author could add a few more explanations as I suggested above.

Reviewer #2 (Remarks to the Author):

In the manuscript of H. Idzuchi et al., a magnetic Josephson junction and the corresponding SQUID are studied, which are composed of van der Waals Ising superconductors and atomically thin ferromagnetic insulators. The system exhibits non-trivial phases for two kinds of critical currents, suggesting two degenerate states that are commonly observed in a ϕ -JJ.

It is quite interesting to construct a van der Waals heterostructure to function as a π - or ϕ -JJ, which is important for building a concrete microscopic model. The idea of combining Ising superconductivity and out-of-plane magnetic insulator is also intriguing. However, I have some concerns about the experimental results.

1. The basic characterization of MJJ is shown for B fields along the in-plane directions in Fig. 2. It's not clear why the Fraunhofer pattern is in lack for the 6 ML device. How about the 1 and 2 ML devices shown in Fig. 1?

To gain more insight into the magnetic domain structure, i.e. bubble or strip, the in-plane fields measurement of Fraunhofer pattern is suggested to be checked with different azimuthal angles.

2. To substantiate the proposal of a ϕ -JJ, I_c versus in-plane B fields should be presented to show the evolution of four critical currents.

3. Line 167, the simulated results are 100 and 260 degrees. Any explanation for the discrepancy with the observed 59 and 259 degrees?

4. In Fig. 3b, a dynamic distribution of critical currents is found in the negative B field. Any reasons for the absence on the positive side? Such a chaos could be observed in a ϕ -JJ with ultralow damping, so it should be suppressed by lowering the quality factor, e.g. by increasing temperatures.

Furthermore, the result in Fig. 3b seems to be from a different device with that in Fig. 3c and d, as they have very different critical currents. Are there more characterization for the device in Fig. 3b to correlate the telegraph signal with two critical currents obtained by different sweeping procedures?

5. In a system of low dissipation, the ground state proves to be randomly trapped at positive or negative ϕ potential well. For instances in the work by H. Sickinger et al (PRL 109, 107002), when the system has a very low damping below 2.3 K, the specific sweeping procedure cannot prepare the ground state anymore. In this regards, the analysis of two critical current in Fig. 4e-f should be carefully treated.

Besides, there are some typos to be corrected. Some of them are listed in the following:

1. Line 38, 'include';
2. Line 42, 'result';
3. Line 96, 'When';
4. Line 145, 'denoted by'

Reply to Reviewers

We appreciate reviewers' positive assessment on the novelty of van der Waals superconducting/ferromagnetic junction. We have carefully considered and implemented all of their helpful and constructive comments and suggestions in the revised manuscript accordingly as describe details below.

Response to Reviewer #1

In this manuscript, Izuchi et al. demonstrated Josephson junction composed of superconductor/ferromagnetic insulator/superconductor (SFS). In this junction, they utilized layered material such as NbSe₂ and CrGeTe to construct van der Waals SFS junction. It is also noticeable the NbSe₂ is having rather unique properties as a superconductor since it is highly anisotropic (Cooper pair spins are aligned to out-of-plane) and noncentrosymmetric in case of few-layer NbSe₂. This system under investigation is novel and could be interesting to explore novel physics originated from superconductor/ferromagnet interface. As the author claimed, use of van der Waals interface can significantly reduce the imperfections in the interface compared to the junction fabricated by a metal deposition technique. The results of this study could be interesting to the community of both superconductor and two-dimensional material sciences. However, I feel that some of the explanations and interpretation of the results are not convincing enough and need to be improved. I would recommend the author to add an explanation for the following points.

Response 1.0: We thank the reviewer for carefully reviewing our paper. We are glad that the reviewer recognizes the scientific significance of our results. We also thank the reviewer for pointing out some missing discussion around the magnetic response, the correction of which has strengthened the paper significantly. We address each point below.

1) What is the thickness of the NbSe₂?

Response 1.1: The thickness of NbSe₂ ranged between 8 nm and 16 nm (on average 12 nm). The only exception is the top flake of the 2ML junction, which is about 100 nm thick. The experimental and theoretical implications of the effect of thickness in our experiment will be discussed in more details in response to reviewer #1's comment "3". The thickness information is now included in the Methods section (the third sentence in "Device Fabrication"):

The thickness of NbSe₂ flakes ranges from 8 nm to 16 nm (on average 12 nm) except for the top flake of the 2ML junction (100 nm) and the NbSe₂/NbSe₂ junction in supplementary Note S2.

We note the data of the NbSe₂/NbSe₂ junction is included in a new supplementary Note S2 in order to respond

to reviewer #2's comment.

2) *The hysteresis observed in the junction Josephson critical current presented in Fig. 2(a,b) looked interesting and may capture something exotic phenomena. However, I am not sure about their interpretation presented in supplementary information.*

(a) *First, the magnetism can change with its size. Therefore, I would imagine that their thin $\text{Cr}_2\text{Ge}_2\text{Te}_6$ can have different saturation fields compared with its bulk form. It may even have hysteresis in thin film. Although authors discuss that hysteresis they observed appears to have a different magnetic field range with the saturation field of bulk $\text{Cr}_2\text{Ge}_2\text{Te}_6$, it may not rule out the possible contribution of the magnetism of CrGeTe material. Please add some explanation or some reference that studies the size dependence of magnetism on CrGeTe(if available).*

Response 1.2a: We thank the reviewer for bringing up this issue. While we do not have a direct magnetization measurement in our atomically thin $\text{Cr}_2\text{Ge}_2\text{Te}_6$ samples, fortunately, there are references that reported the magnetic hysteresis loop in 6ML and slightly thicker ($\geq 8\text{ML}$) samples (Nat **546** 265 and Appl. Phys. Lett. **115** 232403, shown in Fig.R1 below). According to these works, the saturation fields of atomically thin $\text{Cr}_2\text{Ge}_2\text{Te}_6$ samples are similar or smaller than those values found in bulk, and many orders of magnitude different from that of Fig.2a-b ($\sim \pm 1.5 \text{ T} - \sim \pm 2.0 \text{ T}$, for 6ML).

[Redacted]

Figure R1. The magnetic field hysteresis of thin $\text{Cr}_2\text{Ge}_2\text{Te}_6$ flakes from the literature. Left: Nat. 546 265, Right: Appl. Phys. Lett. 115 232403. 6 ML $\sim 4.2 \text{ nm}$.

In the revision, we added this clarification in the main text (6th line of the paragraph starting with “To demonstrate the”) underlined below:

Interestingly, we find that the hysteresis in the magnetic field reaches values of $\sim \pm 1.5 \text{ T}$, much larger than the saturation field (the field required to reach the saturation magnetization) of our $\text{Cr}_2\text{Ge}_2\text{Te}_6$ bulk crystals (Fig. 2c), and that of reported values in bulk crystals and thin flakes [15,18,19].

Related to this point, we have found a typo in Fig.2c: the text $d_f = 6ML$ should be removed since the data is from a bulk sample (as stated in the caption, the main text and Supplementary information). We apologize for this confusion, and have corrected the figure in the revised manuscript.

(b) Secondly, I do not understand why the explanation in Fig. S1 can cause the change of critical current in the SFS junction. Author discusses the change of magnetic domain pattern in CrGeTe due in relation to the vortex formation in NbSe₂. It would be great if the author can explain in more specifically how their explanation increases or decreases the critical current.

Response 1.2b: We appreciate the reviewer suggesting further description of the critical current hysteresis we observe in our SFS junction. Large hysteresis loop in $I_c(H)$ (larger than the hysteresis loop of magnetization) and sudden decrease of $I_c(H)$ near the zero magnetic field, which expects domain reversal (Fig.2c and R1), lead us to speculate the interaction between the magnetic domain in Cr₂Ge₂Te₆ and vortex lattice in NbSe₂. Particularly, forming a stripe magnetic domain pattern near the domain reversal can cost extra energy in vortex configuration in the superconductor. Thus, during the transition from the triangular bubble domain to the more stable stripe domain, one expects a reduction of Josephson free energy, resulting in the step in $I_c(H)$.

In the revision, we added the explanations in the main text (the last sentence of the paragraph starting with “To demonstrate the”):

We conclude that the interplay between the magnetic domains of Cr₂Ge₂Te₆ and the field-dependent Abrikosov vortex lattice in NbSe₂ can induce a transition between magnetic states and explain the experimental observations, including the sudden drop in $I_c(H)$ near zero magnetic field. This critical current drop can be attributed to the differences in the system energy for the two different vortex states, which is interacting underlying magnetic domains (see Fig.S1 and Supplementary Note S1 for detail).

We also added a more detailed explanations in the supplementary materials:

As shown in Fig. S1, one possible scenario to explain our experimental observation (Fig. 2) is to consider the interaction between the magnetic domain in Cr₂Ge₂Te₆ and vortex lattice in NbSe₂. Particularly, forming a stripe magnetic domain pattern in the Cr₂Ge₂Te₆ layer near the domain reversal can cost extra energy in vortex configuration in the superconductor. Thus, during the transition from the triangular bubble domain to the more stable stripe domain, one expects a reduction of Josephson free energy, resulting in the step in $I_c(H)$.

(c) The explanation presented in Fig. S1 does not depend on the junction bias (or junction current) as far as NbSe₂ is superconducting and disappears when NbSe₂ is in a normal state. Does the author have extra data that shows hysteresis disappears when NbSe₂ is in normal states such as applying large current or increasing

measurement temperature?

Response 1.2c: The reviewer correctly pointed out that the hysteresis of the voltage reading in our junction should disappear when NbSe₂ is in the normal state. Indeed, we have the data which shows that the hysteresis disappears when NbSe₂ is in the normal state. As shown in the newly added Fig. S2(R2), hysteresis disappears when a large bias current is applied.

Fig. R2 [new figure in SI as S2]. The field-dependence of the voltage at high bias for the device NbSe₂/Cr₂Ge₂Te₆/NbSe₂. The device is the same as the one shown in Fig. 2a. The thickness of the Cr₂Ge₂Te₆ is 6ML. The temperature is at 0.3 K. The bias current is 0.172 μ A and the field direction is out-of-plane.

In the revision, we added the explanation in the main text (as 7th sentence of the paragraph starting with “To demonstrate the”):

At high bias, which far exceeds the critical current, the voltage across the junction does not show notable hysteresis (Supplementary information, Fig. S2)

We also added Fig. R2 as a new figure S2 in the Supplementary information with a brief description in the last paragraph of Note S1:

To compare with the hysteresis of critical current as a function of the field, we also measured the field dependence of the voltage at a high bias (much larger than the critical current). This measurement did not show notable hysteresis, as shown in Fig. S2.

(d) The vortex formation process in Fig. S1 should be different for the application of an in-plane magnetic field. However, the author observed a similar range of hysteresis in Fig. 2(a) and 2(b) accompanying a similar sudden jump at low field. I wonder how they can explain the behavior on Fig. 2(b) with their model.

Response 1.2d: The initial vortex is introduced as the field reaches the lower critical field B_{c1} . The reported out-of-plane critical field B_{c1} (19 mT) is considerably larger than the in-plane critical field (10 mT) (M. Zehetmayer

and H. W. Weber, Phys. Rev. B **82**, 014524, reference number 44). This is consistent with the hysteresis we measured: when the field is decreased from a large positive value (red curves in Fig. 2a and 2b), the point at which I_c starts to decrease is larger for the out-of-plane sweep. We added this explanation in the Supplementary Note S1 (as the last sentences in the paragraph starting with “The magnetization of bulk $\text{Cr}_2\text{Ge}_2\text{Te}_6$ ”):

Further, in NbSe_2 , the lower critical field B_{c1} is found to be 19 mT and 10 mT for the out-of-plane and in-plane fields, respectively [44]. The low field behavior of Figs. 2a and 2b are attributed to this difference.

3) From their explanation in the method section, I am assuming their NbSe_2 is thick bulk crystal. The multilayer NbSe_2 is having Fermi level at Gamma-point, not only the K-point. Some of the previous study probably suggested that superconductivity possibly appears in both Gamma-point and K-point. Author presented the argument shown below to exclude the contribution from Gamma-point, but I am not sure whether this is really true. Is there any experimental result (could be reference paper) that can justify their assumption that contribution from gamma point is negligible.

“Because of weak interlayer hopping, however, bulk NbSe_2 can be thought of a stack of weakly coupled Ising superconductors with opposite spin polarization in even and odd layers. Tunneling in a Josephson junction predominantly originates from the layer adjacent to the junction and as a result the supercurrent will be carried by ICPs.”

Response 1.3: The reviewer correctly pointed out that there can be a contribution from the Γ point for our sample’s thickness (8 nm -16 nm). We have already described the possible effect of Γ point in the Supplementary information (the last paragraph of Note S2 [S4 in the revised manuscript]). Indeed, as the reviewer suggests, for out-of-plane magnetization, a contribution from K and K’ bands results in a π -coupling whereas the one from Γ band gives in a zero-phase coupling as we described:

When the magnetization is out of plane, our simple model yields no Josephson current because of spin conservation. In that case, other contributions, e.g., from the Γ point or due to spin mixing around the K and K’ point, would presumably lead to a ground state at zero phase. In conclusion, there is a similar competition between zero- and π -junctions that can conceivably result in an overall ground state at a nontrivial phase $\phi \neq 0, \pi$.

While this Γ point contribution may modify the relative amplitudes of π - and 0- couplings, this additional contribution adds a disturbance of the exact values of nontrivial phase values but does not affect the overall interpretation throughout the manuscript. The existence of K/K’ band in a thick sample is experimentally reported in the literature (e.g., Bawden, L. *et al. Nat. Commun.* **7**, 11711, ref. [34] in the previous version of the manuscript, and ref. [32] in the revised manuscript).

In the revision, to refer to the explanation in Supplementary Information, we added a description in the Methods section in “Theoretical model” (5th sentence), underlined below to explicitly pointing the Γ point

effect:

Tunneling in a Josephson junction predominantly originates from the layer adjacent to the junction and as a result the supercurrent will be carried by ICPs. There can be non-vanishing contribution from the Γ point in the Josephson coupling, which modifies the relative amplitudes of π - and 0- couplings of the ICP (see Supplementary information).

4) I believe that the “doubly degenerate ground state” they called appeared Fig. 2(a) and 2(b) as a sudden jump at a low magnetic field, thus can be prepared by magnetic field. In the later section (Fig. 3), they prepared these two states by controlling current. It makes it a bit difficult how to interpolate these states prepared by two different sequences are the same. It should be more simpler if the author prepared the initial state in Fig. 3 with a magnetic field, but they must have some reason not to do that. I recommend the author to add more explanation to explain the relationship between the two experiments. I also wonder why the current sweep can change the initial state.

Response 1.4: We appreciate the reviewers’ very helpful and constructive suggestions made in this comment. As the reviewer correctly pointed out, there are bi-stable magnetic structures in our FJJ that can be switched by a magnetic field as shown in Fig. 2a and 2b. The important difference in Fig. 3 is that the two degenerate critical current states prepared by controlling current are obtained in the same magnetic configuration. The two degenerate states can be identified as the local minimum of the Andreev bound state spectrum (‘washboard potential’) as discussed in Fig. 4d. The Josephson current can drive the transition between these two local minima, which can tilt the washboard potential as illustrated in the inset of Fig. 4f, following the standard description of resistively and capacitively shunted junction (RCSJ) model.

Following the reviewer’s suggestion, we find that it is important to clearly distinguish these two different bi-stabilities (one in magnetic structures controlled by magnetic field and the other from Andreev energy spectrum controlled by Josephson current). To clarify these points, we revised the description in the paragraph of the main text starting with “To probe possibly anomalous Josephson” from:

However, we observe an irregular SQUID response in the field range between -1.2 mT and -2.2 mT, with a telegraph-like signal oscillating between two metastable critical current branches (Fig. 3b) possibly related to the sudden change in critical current seen in Fig. 2a.

To:

However, we observe an irregular SQUID response in the field range between -1.2 mT and -2.2 mT, with a telegraph-like signal oscillating between two metastable critical current branches (Fig. 3b and 3c). This bi-stability is possibly related to the sudden change in critical current seen in Fig. 2a caused by the change of magnetic structure in the junction.

We note that a new Fig.3c is added in order to respond to reviewer #2's comment.

Also, we added the explanation in the paragraph of the main text starting with "A parallel arrangement of 0-":

These two states can be obtained by sweeping back from positive or negative bias as the switching currents can be simply controlled by choosing two metastable states in the bi-stable potential.

Overall, results seem to be new and possibly could promote further discussion in the research community. I would think this can be a good manuscript if the author could add a few more explanations as I suggested above.

We hope to have responded to all the valuable suggestions and strengthened the explanations accordingly. We believe our manuscript is significantly improved. We appreciate Reviewer #1's evaluation of the manuscript.

Response to reviewer #2:

In the manuscript of H. Idzuchi et al., a magnetic Josephson junction and the corresponding SQUID are studied, which are composed of van der Waals Ising superconductors and atomically thin ferromagnetic insulators. The system exhibits non-trivial phases for two kinds of critical currents, suggesting two degenerate states that are commonly observed in a ϕ -JJ.

It is quite interesting to construct a van der Waals heterostructure to function as a π - or ϕ -JJ, which is important for building a concrete microscopic model. The idea of combining Ising superconductivity and out-of-plane magnetic insulator is also intriguing. However, I have some concerns about the experimental results.

Response 2.0: We thank the reviewer for the positive feedback for our combination of the van der Waals superconducting and magnetic structure and Ising superconductivity with an out-of-plane magnetic insulator. We share the opinion that our demonstration can serve as an interesting platform for a concrete microscopic model owing to the high crystallinity and the availability of many interesting materials. In the following, we have addressed the reviewer's concerns and comments, thereby strengthening our work.

1. The basic characterization of MJJ is shown for B fields along the in-plane directions in Fig. 2. It's not clear why the Fraunhofer pattern is in lack for the 6 ML device. How about the 1 and 2 ML devices shown in Fig. 1? To gain more insight into the magnetic domain structure, i.e. bubble or strip, the in-plane fields measurement of Fraunhofer pattern is suggested to be checked with different azimuthal angles.

Response 2.1: We very much appreciate the reviewer's constructive comments and suggestions. Indeed, we have measured the Fraunhofer pattern of some of our F-JJ in the presence of in-plane magnetic fields. Figure R3 below (which is also new Fig. S5 in the revision) summarizes the Fraunhofer pattern obtained from 1ML and 6ML NbSe₂/Cr₂Ge₂Te₆/NbSe₂ Josephson junctions (JJs) and NbSe₂/NbSe₂ JJ without Cr₂Ge₂Te₆ layer as a comparison. Here, the device without Cr₂Ge₂Te₆ was fabricated under ambient conditions, and the thickness of the top and bottom flakes are 12.7 nm and 100 nm, respectively. These JJs exhibit Fraunhofer patterns with multiple peaks as well as the main peak around the center. The second peaks are located at 0.14 T, 0.59 T and 0.79 T (average of the absolute values for positive and negative fields) for NbSe₂/NbSe₂ JJ, NbSe₂/Cr₂Ge₂Te₆(1ML)/NbSe₂ JJ, and NbSe₂/Cr₂Ge₂Te₆(6ML)/NbSe₂ JJ, respectively. In a Fraunhofer consideration, these values correspond to $d+2\lambda$ of 6.8 nm, 6.2 nm, and 13.2 nm, where λ is the penetration length and d the thickness of the barrier layer. These values provide a reasonable agreement, considering the nonideal geometry and the experimental inhomogeneity unavoidable for multiple reasons (e.g., the shape of the junction, the distribution of the current). In addition, irregularity of the curve can result from a field-induced magnetic texture, including a sudden change induced by the domain wall nucleation. Therefore, it is challenging to extract useful information by further quantitative analysis of the Fraunhofer pattern. The reviewer also made

an excellent suggestion of studying the Fraunhofer pattern with azimuthal angular dependence of applied magnetic field to find more direct evidence of magnetic domain structures. While such experiments will indeed enhance our understanding of F-JJ, those experiments require vector magnet control and well-defined device geometry, which we leave for future studies.

Figure R3 [new figure in SI as S5] Fraunhofer pattern. In-plane field response of **a**, NbSe₂/NbSe₂ JJ. **b**, NbSe₂/Cr₂Ge₂Te₆(1ML)/NbSe₂ JJ. **c**, NbSe₂/Cr₂Ge₂Te₆(6ML)/NbSe₂ JJ. The temperature is at 1.6 K, 1.5 K, and 0.3 K from (a) to (c). The inset (green filled area) shows the shape of the junction with B field direction. L_{ave} indicates the average length of the junction along B field direction.

In the revision, reflecting the reviewer's valuable suggestion, we include the Fraunhofer measurement (Fig. S5 above) and the following discussion as Supplementary Note.

Supplementary Note S3. Fraunhofer pattern.

We have also studied the response of the in-plane magnetic field for devices besides the 6-ML one shown in Fig. 2. Figure S5 summarizes the response for NbSe₂/Cr₂Ge₂Te₆/NbSe₂ Josephson junction (JJ) as well as NbSe₂/NbSe₂ JJ made between two cleaved surfaces. We note the device without Cr₂Ge₂Te₆ was fabricated under ambient conditions, and the thickness of the top and bottom flakes are 12.7 nm and 100 nm, respectively. In addition to the clear Fraunhofer pattern in the NbSe₂/NbSe₂ junction, the devices with Cr₂Ge₂Te₆ show multiple peaks with the main peak around zero field. The second peaks are located at 0.14 T, 0.59 T and 0.79 T (averaged for positive and negative fields) for NbSe₂/NbSe₂ JJ, NbSe₂/Cr₂Ge₂Te₆(1ML)/NbSe₂ JJ, and NbSe₂/Cr₂Ge₂Te₆(6ML)/NbSe₂ JJ, respectively. The critical current in a Fraunhofer pattern is given by

$$I_c(X) = I_c(0) \left| \frac{\sin(\pi X)}{\pi X} \right|,$$

with $X = \Phi / \Phi_0$, $\Phi = (d + 2\lambda)LB$, where λ is the penetration length, d the thickness of the barrier layer, L the length of the junction and $\Phi_0 = 20.7 \text{ Gauss} \cdot \mu\text{m}^2$. Inserting these values in the equation above provides with $d + 2\lambda$ values of 6.8 nm, 6.2 nm, and 13.2 nm, which is reasonable values considering the nonideal device

geometry and unavoidable inhomogeneity. Further irregularity can be caused by a field-induced development of magnetic texture, including sudden change due to domain wall nucleation.

2. To substantiate the proposal of a ϕ -JJ, I_c versus in-plane B fields should be presented to show the evolution of four critical currents.

Response 2.2: We thank the reviewer point out the importance of examining four critical current branches in a ϕ -JJ. Such study was done in a long channel metallic F-JJ (refs. 28 and 29 in the revised manuscript). In our device, we also observed four different critical currents (Fig. R4a, new Fig. S3a) that can be tuned by the sweeping procedure and by the out-of-plane magnetic fields (inset to Fig. R4a), which shows some similarity (two branches appear at the side of the minimal point; Fig.5 in ref. 29) with the long channel metallic F-JJ which is tuned by in-plane field. Figure R4b shows that the critical currents of our F-JJ do not show appreciable change with the in-plane magnetic field, unlike long channel metallic F-JJ. These behaviors are consistent with our explanation of Ising-Cooper-pair coupling (Fig.4a and 4b), i.e., our vdW F-JJs are less sensitive to in-plane magnetic fields, rather sensitive to the out-of-plane field, due to the strong spin-orbit coupling of the Ising Cooper pairs.

Figure R4 [new figure in SI as S3] Detail of ϕ -junction behavior. a, Four switching-current branches of ϕ -junction for the device shown in Fig. 3b, under out-of-plane magnetic field. The temperature is at 1.8 K. Inset shows the switching current when sweeping the current from negative values to positive (shown in red) is different from that when sweeping from positive values to zero (not shown) and then from zero to positive values (black). The magnetic field is at -2.2 mT and the temperature is at 2.1 K. **b**, switching currents under the in-plane magnetic field at the temperature of 1.4 K. Inset shows the ones for larger field range. Note that the background tilt in I/V curve is removed.

In the revision, reflecting the reviewer’s helpful suggestion, we include the four critical currents (Fig. S3 above) and the following discussion as a Supplementary Note:

Supplementary Note S2. Further characteristics of the SQUID with a 1ML $\text{Cr}_2\text{Ge}_2\text{Te}_6$ barrier junction.

In this section, we provide further characteristics of ϕ -junction. We studied the device of Fig. 3b after

removing a part of the link in the arm in the SQUID connection. This allows us to see two-critical currents without SQUID oscillations. We have confirmed the device shows two different critical currents with one bias polarity, one for each sweep sequence (inset of Fig. S3a). Four branches for switching current were characterized by varying current sweep sequences for both positive and negative bias directions (Fig. S3a). Figure S3b shows the critical current under the in-plane field.

3. Line 167, the simulated results are 100 and 260 degrees. Any explanation for the discrepancy with the observed 59 and 259 degrees?

Response 2.3: Our theory employs a simplified model. We acknowledge those phase values in Fig. 3d can deviate from the values calculated in this simplified model. Further deviations can result from shape anisotropy and inhomogeneity of the current in experimentally realistic devices (some of these factors were discussed in Response 2.1 above). We added the explanation in the main text, as the last sentence in the paragraph starting with “A parallel arrangement of 0-”:

We also note that the experimental value of ϕ_1 deviates substantially from the theoretical value obtained above. This can be attributed to the experimental anisotropy and inhomogeneity of the devices as observed in the in-plane field Fraunhofer pattern (see Supplementary Note S3), which is not considered in our theoretical analysis above.

4. In Fig. 3b, a dynamic distribution of critical currents is found in the negative B field. Any reasons for the absence on the positive side? Such a chaos could be observed in a phi-JJ with ultralow damping, so it should be suppressed by lowering the quality factor, e.g. by increasing temperatures. Furthermore, the result in Fig. 3b seems to be from a different device with that in Fig. 3c and d, as they have very different critical currents. Are there more characterization for the device in Fig. 3b to correlate the telegraph signal with two critical currents obtained by different sweeping procedures?

Response 2.4: The reviewer correctly pointed out that the SQUID critical current (Fig. 3b) exhibits a bi-stable critical current that display ‘telegraph’ type of two-state fluctuation in the magnetic field regime around -1.5 mT, and also the result of Fig. 3b is from a different device with that in Fig. 3c and 3d (Fig. 3d and 3e in the revised manuscript). Indeed, as the reviewer noted, the data in Fig. 3b is obtained from a different device from the one we showed the data in Fig. 3c and 3d (Fig. 3d and 3e in the revised manuscript) to clearly show the field-induced bistability. We appreciate the reviewer’s suggestion to add more characterization of the device in Fig. 3b to correlate the telegraph signal with the characteristics in the device in Fig. 3c and 3d (Fig. 3d and 3e in the revised manuscript). While Fig. 3b may have shown some ambiguity whether the signal around -1.5 mT indicates indeed two levels, now as shown in new Fig. 3c [added Figure in this revision], the top and bottom

states are more clearly representing two levels which smoothly connects to the outer region of the magnetic field. Therefore, the device Fig. 3b shows a strong resemblance of the behaviors shown in phi JJ devices in Fig. 3c and 3d (Fig. 3d and 3e in the revised manuscript). We also note that the two branches can be prepared by different sweeping procedures (inset of Fig. R4a).

Also, we thank the referee for raising the important issue of the field polarity. As shown in Fig. 4, our phi-JJ is formed due to the magnetic texture, and we believe its formation (including a chirality of domain wall) would depend on the polarity of both the magnetic field from the electromagnet and the current-induced field.

Finally, we thank the reviewer’s suggestion of raising the temperature to lower the quality factor of the F-JJ to reduce chaotic behavior. We indeed have temperature-dependent measurement results of two critical currents which shows suppression of the chaotic behavior with increased temperature (Fig. R5).

Fig. 3c [new figure in main text] The detailed view of the switching region of bistable SQUID critical current oscillations marked by dashed box in (b).

Fig. R5 [new figure in SI as S4] Switching probability P of ϕ -JJ with different temperatures at the magnetic field of -4.3 mT, where I_0 denotes the bias current at $P = 0.5$. The bias sweep is given from large positive through zero to large positive.

Following the reviewer’s careful comment, we revise as follows. First, we include the detailed view of ‘telegraph signal’ (new Fig. 3c) above and refers to it in the paragraph of the main text starting with “To probe possibly anomalous Josephson”:

However, we observe an irregular SQUID response in the field range between -1.2 mT and -2.2 mT, with a telegraph-like signal oscillating between two metastable critical current branches (Fig. 3b and 3c).

Second, we added the explanation in the paragraph of the main text starting with “A parallel arrangement of 0-”:

The telegram-like signal in Fig. 3b is found in the negative magnetic field region. A careful examination of this two-state switching behavior implies that two meta-stable SQUID oscillations with different phases involve (Fig. 3c), suggesting that the bistability can also be controlled by applied magnetic fields.

Third, we added the explanation in the revised figure caption:

c, The detailed view of the switching region of bistable SQUID critical current oscillations marked by dashed box in (b). d, The sweep-dependent critical currents for Cr₂Ge₂Te₆-SQUID measured at T = 1.1 K. The SQUID device is similar to the one shown in (b) but with a smaller critical current.

Finally, we include the temperature variation of switching probability (Fig. S4 above) and the following discussion in the second paragraph of Supplementary Note S2:

The probability of switching current was also characterized. Figure S4 shows the temperature variation at the field of -4.3 mT, which shows two switching currents (Fig. S3b), indicating suppression of stochastic behavior with raising the temperature.

5. In a system of low dissipation, the ground state proves to be randomly trapped at positive or negative ϕ potential well. For instances in the work by H. Sickinger et al (PRL 109, 107002), when the system has a very low damping below 2.3 K, the specific sweeping procedure cannot prepare the ground state anymore. In this regards, the analysis of two critical current in Fig. 4e-f should be carefully treated.

Response 2.5: Indeed, determinism in ϕ -junction is an interesting topic and discussed in H. Sickinger et al. (ref [29] in the revised manuscript and ref [31] in the original manuscript). In the plot of escape rate as a function of the bias current (Fig. 4f), we can convert the tilt to corresponding energy. The dashed lines in Fig. 4f show that the energy is much higher than the environment temperature (1.1 K), which is consistent with the framework of the analysis we have employed.

In the revision, following the suggestion of the reviewer, we added this information in the caption of Fig.4f:

Escape rate τ^{-1} as a function of the bias current that has a different slope (corresponding to an energy higher than the environment temperature) reflected by the different metastable states of the tilted washboard potential shown in **d**, and the potential barrier for each escaping event is schematically illustrated in the insets.

Besides, there are some typos to be corrected. Some of them are listed in the following:

1.Line 38, 'include';

2.Line 42, 'result';

3.Line 96, 'When';

4.Line 145, 'denoted by'

Response 2.6: We appreciate the reviewer's careful reading to find these typos. We also sincerely apologize for the typos and for mislabeling ($d_F = 6\text{ML}$) in Fig.2c (it is the bulk sample data, as we clearly stated in the caption, main text, and supplementary information). In the revised manuscript, we have corrected all of the typos mentioned above and further improved the language in other parts of the text.

REVIEWERS' COMMENTS

Reviewer #1 (Remarks to the Author):

I believe that author carefully considered all the criticism raised by reviewers and correctly revised the manuscript. Technically, I do not see any problem in the revised version of the manuscript. Since they presented novel type of Josephson coupling in van der waals junction, the manuscript could be suitable for submitted journal.

Response to Reviewer

Response to Reviewer #1

I believe that author carefully considered all the criticism raised by reviewers and correctly revised the manuscript. Technically, I do not see any problem in the revised version of the manuscript. Since they presented novel type of Josephson coupling in van der waals junction, the manuscript could be suitable for submitted journal.

Response: We thank the reviewer for the positive recommendation on our paper and our responses.